# The Effect of SSRI Exposure in Pregnancy on Early Respiratory and Metabolic Adaptation in Infants Born Preterm

**DOI:** 10.3390/children10030508

**Published:** 2023-03-04

**Authors:** Ayala Gover, Kareen Endrawes, Michal Molad, Karen Lavie-Nevo, Arieh Riskin

**Affiliations:** 1Neonatal Intensive Care Unit, Bnai-Zion Medical Center, Haifa 3339419, Israel; 2Rappaport Faculty of Medicine, Technion, Israel Institute of Technology, Haifa 3525433, Israel; 3Neonatal Intensive Care Unit, Lady Davis Carmel Medical Center, Haifa 3436212, Israel

**Keywords:** SSRI, preterm, early neonatal adaptation

## Abstract

Selective serotonin reuptake inhibitors (SSRIs) are increasingly used for maternal depression during pregnancy; however, their use has been linked to adverse effects in newborns. Respiratory and feeding problems, jaundice, metabolic and temperature dysregulation and hypoglycemia have been described in term infants. However, scarce data exists on early neonatal adaptation in exposed infants born prematurely. We aimed to assess the effects of SSRI exposure on early neonatal adaptation measures in infants born prematurely. Data from preterm infants exposed to maternal SSRIs during pregnancy and from matched controls were retrospectively collected. Forty-two infants comprised the final cohort: 21 infants with SSRI exposure and 21 matched controls. 1 min Apgar score was significantly lower in the exposed group compared to the non-exposed group (*p* = 0.043). No differences were found in 5 min Apgar scores, cord pH, need for delivery room resuscitation, rate of hypoglycemia, hyponatremia, hyperbilirubinemia, need for phototherapy, temperature stability and maximal oxygen requirements. No differences were found in the total time of respiratory support, time to reaching full enteral feeds, length of stay and complications of prematurity. Unlike studies in term infants, no significant differences were found in adaptation and short-term outcomes between preterm infants with and without SSRI exposure in pregnancy.

## 1. Introduction

Mood disturbances such as depression and anxiety occur in up to 20% of pregnant women [1,2] and are serious conditions associated with various adverse maternal and neonatal outcomes if left untreated [2,3,4]. Maternal depression during pregnancy has been associated with preterm birth and low birth weight [4,5], impaired neurobehavioral development [4], increased risk of post-partum depression and impaired maternal–infant bonding [6]. The use of antidepressants in pregnancy has grown over the years to an estimated rate of 6–13% of pregnant women [1,3]. Selective serotonin reuptake inhibitors (SSRIs) are frequently chosen due to their perceived high effectiveness and relatively favorable safety profile [7]. However, substantial placental passage and fetal exposure has been established [8], and their use in late pregnancy has been linked to adverse effects in the newborn.

An increased risk of preterm birth has been described across studies [9]. One systematic review and meta-analysis of eight studies from North America and Northern Europe [6] with a total of 1,237,669 women compared the incidence of preterm delivery in women with depression, with and without use of SSRIs during pregnancy, with a control group of women with no depression and no drug use. The risk of preterm labor was significantly higher in the women with depression who were treated with SSRIs compared to controls and to women with depression without treatment with SSRIs, even after adjusting for various confounders. Birth weight was significantly lower in the exposed group, and the rate of respiratory distress syndrome was higher. Five of the eight studies included women who received SSRIs only in the first trimester. Women who received SSRIs during the third trimester had a significantly higher risk of preterm birth compared to treatment in the first trimester.

In utero exposure to SSRIs has been associated with poor neonatal adaptation, thought to be related either to serotonergic overstimulation or to withdrawal phenomena after birth [10,11]. One prospective study in term infants found a rate of 30% of symptoms consistent with neonatal abstinence syndrome (NAS), evaluated by the Finnegan score in the group of exposed newborns (n = 60), compared to none in the healthy control group (n = 60) [12]. Thirteen percent of exposed infants had severe NAS (Finnegan score ≥ 8) and seventeen percent had mild NAS (Finnegan score 4–7). Hypoglycemia was noted in 5% of the exposed infants. Another retrospective study utilizing a modified Finnegan score in 220 term and preterm infants exposed to SSRIs in the third trimester reported a rate of 3% of severe abstinence syndrome and 22% of mild abstinence [13]. Hypoglycemia was common (19%). No differences in outcomes were found between different SSRIs (citalopram, sertraline, fluoxetine or others). Only 19/220 newborns in this cohort were born preterm. Interestingly, time to maximal Finnegan score reflecting peak abstinence symptoms ranged between 2 and 90 h; therefore, term infants discharged from the hospital earlier might not have been captured. Another study comparing behavioral and other manifestations, such as tone abnormalities, respiratory symptoms and jaundice, in 76 exposed versus 90 non-exposed newborns found a rate of 77% of abnormal findings in the exposed group compared to 41% of unexposed newborns (*p* < 0.001) [14]. The symptoms resolved within 5 days. In the same cohort, a comparison between 21 exposed preterm infants versus 55 exposed term infants revealed significantly more abnormal findings in the preterm group compared to the term group (100% vs. 69% respectively, *p* = 0.002) [14]. However, as muscle tone, tachypnea, apneas, hypoglycemia and jaundice were some of the evaluated parameters, this difference may be at least partially attributed to effects of prematurity.

Most available studies on early adaptation and respiratory and metabolic regulation were performed on term infants, while some included a mixed population of term and preterm infants, with the majority of the cohort being term infants. Less data are found exclusively on preterm infants. The purpose of this study was to assess the effect of in-utero SSRI exposure on early adaptation and short-term outcomes in infants born preterm. We hypothesized that SSRI exposure would be associated with poor respiratory and metabolic adaptation in preterm infants.

## 2. Materials and Methods

This was a retrospective observational study performed in the neonatal intensive care unit (NICU) at the Carmel Medical Center. Preterm infants with prenatal SSRI exposure and matched controls without SSRI exposure were included. The study was approved by the Carmel Medical Center Research Ethics Board.

Electronic records of all infants born during the years 2015–2022 and admitted to NICU were screened, and all preterm infants with a positive history of maternal SSRI use throughout the pregnancy were included. Controls were matched by gestational age, birth weight, sex and date of birth and a confirmed negative history of maternal SSRIs. Infants with culture-positive early-onset sepsis, maternal diabetes, major congenital anomalies or a highly suspected genetic syndrome and infants who did not survive the first week of life were not included. Infants with unknown maternal drug history, additional psychotropic medications or those in whom maternal SSRI use was discontinued before the third trimester were also excluded. Data were retrospectively collected from an electronic health record database (Metavision^®^, iMDsoft, Park Atidim, Tel Aviv, Israel). Records were reviewed using MetaVision’s Query Wizard tool, identifying infants admitted to NICU during the study period with completed data on maternal drug use during pregnancy. All identified infant records were then reviewed to select the preterm infants exposed to an SSRI and the matched controls.

Retrieved information included:(a)Maternal demographics and health status, pregnancy-related complications and use of prenatal steroids;(b)Early-onset sepsis risk factors (maternal positive GBS swab, prolonged rupture of membranes and maternal fever), gestational age, birth weight and sex of the newborn, need for advanced delivery room resuscitation (advanced airway or chest compressions), cord pH and Apgar scores;(c)Early respiratory and metabolic neonatal adaptation parameters including hypoglycemia episodes, need for hypoglycemia treatment, body temperature regulation, need for respiratory support, need for surfactant treatment, need for inotropes, highest bilirubin level, need for phototherapy, serum sodium levels and time to regaining birth weight;(d)Complications of prematurity such as intraventricular hemorrhage, patent ductus arteriosus, late-onset sepsis, bronchopulmonary dysplasia, necrotizing enterocolitis and retinopathy of prematurity, defined by commonly used published criteria [15,16,17].

Data were recorded in an Excel spreadsheet (Microsoft Office, Seattle, WA, USA) and statistically analyzed using SigmaPlot, version 11.0 (Systat Software Inc., San Jose, CA, USA) and Minitab^®^, version 16.2.2 (Minitab Inc., State College, PA, USA & Coventry, UK). The sample size calculation assumed a 29% baseline rate of post-delivery respiratory distress in an unexposed population consisting mostly of late-preterm newborn infants [18]. Based on an increase of 1.8-fold in respiratory distress in term newborn infants exposed to an SSRI [19], we assumed an increase of 2.7-fold (1.8 × 1.5) in respiratory distress in exposed newborns who were also preterm infants. Thus, the sample size calculation based on an increase from 29% to 78% respiratory distress in the exposed (compared to the unexposed) preterm newborns, with 80% desired power and an alpha of 5% resulted in a minimum of 19 infants in each of the study and control groups. Thus, we assumed that a retrospective study of 8 years would suffice to have enough infants included with full prenatal maternal medication data in each group. The statistical analysis included descriptive statistics, Student’s t-test or the Mann–Whitney (Rank Sum) U-Test for the comparison of continuous variables according to their distributions, and the Chi-Square test or Fisher’s exact test for comparisons of categorical variables. Results are presented as the mean ± standard deviation (SD) and/or median and interquartile range (IQR). *p*-values of less than 0.05 were considered statistically significant.

## 3. Results

During the years 2015–2022, there were 826 infants born at the Carmel Medical Center who were admitted to the NICU and had completed data on maternal drug use in the department’s database. Fifty-two of the infants had a confirmed SSRI exposure during pregnancy. We excluded 29 exposed infants who were born at term, one preterm infant who had only first-trimester exposure and one preterm who had co-exposure to benzodiazepines. Twenty-one preterm infants with SSRI exposure throughout the pregnancy and twenty-one matched controls comprised the final cohort (n = 42). The baseline maternal and neonatal characteristics are presented in Table 1. The range of gestational age at birth was from 26.4 to 35.6 weeks of gestation. In total, ten infants (five in each group) were born below 30 weeks and eleven were born with a birth weight of less than 1500 g. In the SSRI exposure group, ten preterm infants were exposed to escitalopram, five were exposed to sertraline, four to fluoxetine, and two to paroxetine.

Due to matching the controls, the gestational age and birth weight were comparable between the exposed and non-exposed group, but no significant differences were found in other baseline parameters as well. Most preterm infants were appropriate for their gestational age.

### 3.1. Early Neonatal Adaptaion

#### 3.1.1. Delivery Room Parameters

There was a statistically significant difference between the exposed and non-exposed groups in 1 min Apgar score (median (IQR); 8.0 (5.7, 9) and 9.0 (8, 9) respectively, *p* = 0.043). Seven of the exposed infants had an Apgar score below 7 at 1 min (33.3%) versus four of the non-exposed infants (19%). No significant differences were found in the 5 min Apgar scores, cord pH or the need for advanced delivery room resuscitation. (Table 2).

#### 3.1.2. Early Respiratory Adaptation

The need for any respiratory support in the first 3 days of life was comparable between the groups, as was the need for intubation and mechanical ventilation during the first week of life, the need for surfactant treatment and maximal oxygen requirements (Table 2). None of the infants in our cohort developed persistant pulmonary hypertension.

#### 3.1.3. Early Metabolic Adaptation

Blood glucose level and hypoglycemia episodes in the first 3 days of life were comparable between the groups, as were maximal bilirubin levels and the need for phototherapy. The mean lowest serum sodium levels during the first week of life were similar between the groups, although hyponatremia <130 meq/L occurred more often in the exposed group compared to the non-exposed group (19% vs. 4.7% respectively, *p* = 0.345). Bilirubin levels and need for phototherapy were similar.

### 3.2. Short Term Outcomes and Complications of Prematurity

No differences were found in the short-term outcomes or in the rate of complications of prematurity (Table 3). The length of stay, time to reaching full enteral feeds and time to regaining birth weight were comparable between the groups. One infant in each group required the medical closure of a patent ductus arteriosus, and none required surgical treatment. One preterm infant in the SSRI exposure group developed retinopathy of prematurity and was treated with laser photocoagulation. None of the preterms in our cohort were diagnosed with necrotizing enterocolitis or spontanous intestinal perforation, and none had culture-positive late-onset sepsis. Three preterm infants received a blood transfusion during the first week of life: two from the exposed group and one from the non-exposed group. The rate of intraventricular hemorrhage (grades I and II) was lower in the exposed group compared to the non-exposed group (4% vs. 23.8% *p* = 0.077), but this difference did not reach statistical significance.

## 4. Discussion

SSRIs increase serotonin levels by inhibiting its reuptake at the serotonin transporter at the presynaptic neuron. Their use in pregnancy has been increasing in recent years due to their relatively fewer side effects compared to other antidepressants [7,13], and rates of use by up to 15% of pregnant women have been reported [20]. The use of SSRIs in early pregnancy has not been associated with major structural anomalies in most reports; however, some minor defects were described [1,14], and a small increase in the risk for cardiac defects was noted specifically with paroxetine use, although not consistently [21,22].

SSRIs cross the placenta and were shown to be present in amniotic fluid, in cord blood in correlation with maternal dose and to alter serotonin levels in the fetus [2,8]. By crossing the blood–brain barrier, SSRIs may induce high levels of serotonin in the fetal brain as well as acute shifts in serotonin levels during a vulnerable period of brain development. Many early disturbances have been described in exposed neonates, including an increased risk of prematurity and a low birth weight, low Apgar scores, increased NICU admissions, SSRI discontinuation syndrome and autonomic dysregulation, hypoglycemia, respiratory distress syndrome, persistent pulmonary hypertension, feeding problems and an increased length of hospital stay [1,2,10]. Changes in acute pain response [23] and white and gray matter microstructural changes [3] have also been described, as well as long-term neurodevelopmental and behavioral effects lasting into childhood [2,24].

Over the past two decades, the widespread use of SSRIs has prompted extensive research to assess the health risks associated with in utero exposure. However, inherent methodological challenges exist in this field. Isolating the drug effect from the effects of maternal depression and genetic and epigenetic factors is difficult [2]. Even when exposure to depression with SSRIs is compared to exposure to depression without SSRIs, selection bias by indication exists, as treated women may have initiated treatment due to a more severe depression or anxiety state than untreated women. The observational design of the studies does not allow for higher quality evidence; however, the target population and target treatment are unlikely to be studied with randomized allocation [10]. High heterogeneity in populations and in exposure is evident across studies. Pharmacokinetic differences exist between different SSRIs [8], and dosing is highly variable. For example, sertraline dosing may vary from 20 mg/day to greater than 150 mg/day [7], and the length of exposure may vary from as long as 40 weeks of pregnancy to as little as 2 weeks prior to delivery [7,14], thus limiting conclusive results of meta-analyses [25]. In addition, many reports have pooled together exposed term and preterm infants in comparison to unexposed controls [9,10,13,14,26], although prematurity itself is an important confounder in relation to some of the reported outcomes [9]. In most studies including both term and preterm infants, term infants comprise the vast majority of the cohort [13,14,19]. The purpose of this study was to assess early neonatal outcomes specifically in premature infants exposed to an SSRI throughout the pregnancy in comparison to non-exposed preterm controls.

### 4.1. Delivery Room Parameters

In our cohort, the 1 min Apgar score was significantly lower in exposed versus non-exposed infants, in agreement with previous large meta-analyses. In a systematic review and meta-analysis including 33 studies with over 25,000 cases of in utero antidepressant exposure, increased odds ratios for an Apgar score below 7 were found for both 1 and 5 min Apgar scores in exposed infants [10] compared to non-exposed controls. However, newborns born to depressed mothers without antidepressant treatment had comparable odds ratios for low Apgar scores. In another large meta-analysis examining the effects of depression and antidepressant use in pregnancy, low Apgar scores were found particularly with SSRI use (fluoxetine and sertraline) compared to other antidepressant drugs [27]. In preterm infants, one retrospective study comparing 51 exposed preterm infants from 28 to 36 weeks of gestation with 50 non-exposed matched controls [7] found that a 1 min Apgar score < 7 was more frequent in the exposed group, and lower Apgar scores were associated with a higher sertraline dose.

Advanced delivery room resuscitation was chosen in our study to reflect the need for a higher level of intervention beyond the first steps of newborn resuscitation, which are commonly required in the healthy preterm population. This outcome was similar between the groups in our cohort (14% in exposed vs. 9.5% in non- exposed infants, *p* = 1.0). In another study including preterm infants, oxygen administration in the delivery room was slightly higher in exposed versus non-exposed newborns (55% vs. 36%, adjusted *p*-value = 0.018), but the need for positive pressure respiratory support was comparable (22% vs. 14%, *p* = 0.85) [7]. Conversely, in a study on term infants, significantly more newborns required delivery room respiratory support and admission to the NICU with exposure to an SSRI compared to non-exposed newborns (12.9% vs. 4.2%, respectively; unadjusted OR, 3.34; 95% CI, 2.02–5.29) [28].

Overall, since the 5 min Apgar score, cord pH and need for advanced resuscitation were not influenced by SSRI exposure in our cohort, the 1 min Apgar score likely reflects an initial mild transient depression.

### 4.2. Early Respiratory Adaptation

Unlike most studies in term infants, we found no significant differences between the exposed and non-exposed groups in need of respiratory support and in need for surfactant treatment indicating respiratory distress syndrome during the first week of life, and we had no cases of persistent pulmonary hypertension. One study found a significantly higher rate of respiratory distress, mostly transient tachypnea of the newborn, in SSRI-exposed term infants (n = 401) compared to non-exposed, matched controls [29]. However, only 3% of the infants required oxygen for more than 6 h. In a large study using population-based, linked health data [19] including mostly term infants, increased risk of respiratory distress in exposed infants versus non-exposed infants to depressed mothers remained significant after controlling for the severity of maternal depression by using propensity score matching. Our results are in agreement with another study in preterm infants in which positive pressure ventilation and mechanical ventilation were similar in exposed and non-exposed preterm infants [7]. An increased risk of persistent pulmonary hypertension was previously described with SSRI exposure [4,30,31,32], but the absolute risk was small, so it is unlikely to be observed in small cohorts.

### 4.3. Early Metabolic Adaptation

Contrary to reports on term infants, we did not find significant differences between the groups in rates of hypoglycemia or jaundice. Rate of hypoglycemia with exposure to SSRIs in previous reports varied between 3.5 and 19% [33]; however, many of the studies included a mixed term and preterm population and neither consistently accounted for this potential confounder [13,33], nor for maternal gestational diabetes. Increased risk of jaundice was suggested by some studies. However, in a large meta-analysis of 33 studies, rates of jaundice were not affected by SSRI exposure [10], and in another large study, differences in rates of jaundice did not remain significant when maternal disease severity was accounted for using a propensity score [19].

### 4.4. Other Short-Term Outcomes

The length of stay in the NICU was comparable between the groups in our cohort, in agreement with two other studies in preterm infants [7,34]. In one study, which included both term and preterm infants [14], exposed infants had longer lengths of stay; however, prematurity was the only variable increasing the risk for NICU admissions, and the length of stay for patients admitted to the NICU was not different between exposed and non-exposed infants. In a large, population-based study [19], the length of stay was significantly longer in exposed infants in the initial analysis, but this difference was lost in further analysis using propensity score matching for maternal disease severity.

We found no association between SSRI exposure and complications of prematurity, in agreement with another cohort of preterm infants [34] (exposed n = 19, controls n = 19); however, the sample size was relatively small relative to the incidence of most complications. Interestingly, a study in preterm infants examining the effect of SSRI exposure on neurodevelopmental outcomes at 36 months of age found no developmental abnormalities above the baseline risks of prematurity [34].

We hypothesized that in utero exposure to SSRIs would have a more profound effect on preterm infants compared to term infants as they are the most vulnerable and immature population of newborns. However, unlike previous studies in term infants, our results showed no major differences in outcomes in exposed compared to non-exposed preterm infants. Several explanations are possible. First, our sample size was relatively small, and this is a major limitation to the interpretation of our results. Our sample was probably also underpowered to detect differences because some of the assumptions entered in the sample size calculation turned to be wrong. We used respiratory support in the first 3 days of life as a proxy to respiratory distress. Although the rate of post-delivery respiratory support in the study group exposed to an SSRI was close to our assumption (76% vs. 78%), the baseline rate of respiratory support in the unexposed control group was much higher than our assumptions (66% vs. 29%), probably reflecting the wide use, nowadays, of post-delivery non-invasive respiratory support, especially in preterm infants and including late preterm newborns, who were indeed the majority of our study population. Second, some methodological differences exist between our study and previous works as we aimed to avoid the bias of a mixed population of term and preterm infants by focusing exclusively on preterm infants and kept homogeneity in the length of exposure to SSRIs in the study population. In retrospect, it might have been useful to add another control group of term newborns exposed to a maternal SSRI to verify the effects of a maternal SSRI in our population of term infants, although there are plenty of studies from our country on term newborns exposed to a maternal SSRI [12,24,29,30]. Third, exposed infants who are born preterm “miss” the last few weeks of pregnancy so that the total length of exposure to the drug is shorter compared to infants born at term, possibly affecting neonatal outcomes [35]. Furthermore, animal data demonstrated that the fetal drug metabolic capacity increases in late pregnancy, resulting in elevated levels of the biologically active metabolite in the fetus in late pregnancy compared to an earlier stage [36]. Additionally, temporal changes in the pharmacokinetic properties of SSRIs have been shown in pregnant women due to variations in the activity of cytochrome P450 enzymes, maternal weight and blood volume and creatinine clearance along the course of pregnancy, affecting changes in maternal drug metabolism [37].

We were not able to collect Finnegan scores because it was not done routinely in preterm infants in our department. Nonetheless, this score was designed for neonatal abstinence syndrome after opiate exposure and has not been validated for poor neonatal adaptation after SSRI exposure [10]. Other limitations of our study beyond sample size include its retrospective design, lack of dosing information and lack of another control group of preterm infants with maternal depression without SSRI treatment.

## 5. Conclusions

In conclusion, 1 min Apgar scores were lower in preterm infants exposed to an SSRI during pregnancy compared to non-exposed matched controls. Other adaptation parameters and short-term outcomes were similar between the groups. Larger prospective studies are needed to confirm our results.

## Figures and Tables

**Table 1 children-10-00508-t001:** Baseline maternal and neonatal characteristics.

Parameter	SSRI Exposure n = 21	No SSRI Exposure n = 21	*p*-Value
Mode of delivery n (%)			1.0
Vaginal	6 (28.5)	7 (33.3)	
Cesarean section	15 (71.4)	14 (66.6)	
Multiple pregnancy n (%)			0.093
Singleton	15 (71.4)	20 (95.2)	
Twin	6 (28.5)	1 (4.7)	
Prenatal steroids n (%)	11 (52.3)	17 (80.9)	0.102
Risk factors for early neonatal sepsis * n (%)	5 (23.8)	9 (42.8)	0.326
Maternal hypertension or preeclampsia n (%)	5 (23.8)	2 (9.5)	0.410
Sex n (%)			0.755
Male	12 (57.1)	12 (57.1)	
Female	9 (42.8)	9 (42.8)	
Gestational age *** weeks	34.4 (30.6, 34.9)	34.1 (30.7, 34.9)	0.980
Birth weight ** g	1938 ± 745	1945 ± 636	0.974
Weight for gestational age n (%)			0.834
SGA	2 (9.5)	1 (4.7)	
AGA	17 (80.9)	18 (85.7)	
LGA	2 (9.5)	2 (9.5)	
Head circumference ** cm	29.8 ± 3.4	29.9 ± 2.7	0.951

* Risk factors for early sepsis included maternal fever >38 °C, prolonged rupture of membranes > 18 h, positive maternal Group B Streptococcus swab. ** Mean ± SD. *** Median (IQR 25%, 75%). AGA—appropriate for gestational age; SGA—small for gestational age; LGA—large for gestational age.

**Table 2 children-10-00508-t002:** Early respiratory and metabolic adaptation.

Parameter	SSRI Exposure n = 21	No SSRI Exposure n = 21	*p*-Value
1 min Apgar **	8 (5.7, 9.0)	9 (8.0, 9.0)	0.043
5 min Apgar **	9 (9.0, 10.0)	10 (9.0, 10.0)	0.120
Cord pH **	7.34 (7.29, 7.38)	7.33 (7.3, 7.35)	0.791
Delivery room advanced resuscitation n (%)	3 (14.2)	2 (9.5)	1.0
Respiratory support—first 3 days of life n (%)	16 (76.2)	14 (66.6)	0.541
Mechanical ventilation—first week of life, n (%)	6 (28.5)	4 (19.0)	0.717
Surfactant treatment n (%)	7 (33.3)	9 (42.8)	0.751
Maximal FiO_2_ requirements—first week of life * (FiO_2_%)	25.7 ± 3.6	28 ± 2.9	0.102
Maximal bilirubin ** (mg/dL)	11.6 (10.5, 12.5)	11.2 (9.6, 12.7)	0.624
Phototherapy * (days)	2.0 ± 1.4	1.4 ± 1.2	0.896
Lowest blood glucose—first week of life * (mg/dL)	56 ± 15	53 ± 13.3	0.568
Need for hypoglycemia treatment n (%)	8 (38.0)	12 (57.1)	0.354
Lowest serum sodium in serum—first week of life *	136 ± 5.5	136 ± 4.4	0.978
Serum sodium < 130 mg/dL n (%)	4 (19.0)	1 (4.7)	0.345
Maximal body temperature—first week of life **	37.4 (37.4, 37.5)	37.4 (37.2, 37.5)	0.622

* Mean ± SD. ** Median (IQR 25%, 75%). FiO_2_—fraction of inspired oxygen.

**Table 3 children-10-00508-t003:** Short-term outcomes and complications of prematurity.

Parameter	SSRI Exposure n = 21	No SSRI Exposure n = 21	*p*-Value
Length of stay ** (days)	21 (9.5, 49.5)	16 (8.5, 45.0)	0.371
Time off positive pressure ventilation ** (days)	2.5 (1.0, 6.5)	4.0 (2.0, 7.5)	0.403
Time off supplemental oxygen ** (days)	3 (2.0, 35.2)	7.5 (2.0, 22.0)	0.623
Time to full enteral feeds ** (days)	5 (2.0, 8.2)	5 (2.0, 7.2)	0.840
Time to regaining birth weight ** (days)	8 (6.7, 9.0)	9 (5.7, 14.0)	0.340
Antibiotic treatment-first week of life * (days)	2.8 ± 4.4	1.4 ± 1.8	0.229
Number of sepsis workups **	1 (1.0, 2.25)	1 (1.0, 2.0)	0.654
BPD n (%)	5 (23.8)	3 (14.2)	0.695
IVH n (%)	1 (4.7)	5 *(* 23.8 *)*	0.077

* Mean ± SD. ** Median (IQR 25%, 75%). BPD—Bronchopulmonary dysplasia; IVH—intraventricular hemorrhage.

## Data Availability

The data presented in this study are available on request from the corresponding author. The data are not publicly available due to restrictions of privacy.

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
