# Peer review of "The Effect of SSRI Exposure in Pregnancy on Early Respiratory and Metabolic Adaptation in Infants Born Preterm"

_children, 2023, doi:10.3390/children10030508_

Round 1

Reviewer 1 Report

This paper reported the result of a retrospective analysis, aiming to provide data for the relationship between SSRI exposure in Pregnancy and preterm newborn health status.

Overall, the paper was well written. The authors provided respectable information for background, detailed described the method. They also included an exhaustive discussion section comparing their results to previous published data, and brought up possible reasons for the differences.

The authors mentioned in discussion, and I fully agree that the most significant limitations of this paper are the small sample size and being retrospectively designed, which obviously affect the overall value of the paper. Some of the baseline characteristics and results are too small for statistical analysis. Due to this reason, I suggest removing the "significantly" word in line 314. 

Author Response

Thank you for your comments. We greatly appreciate the feedback and have made the appropriate adjustments.

Point 1 - The authors mentioned in discussion, and I fully agree that the most significant limitations of this paper are the small sample size and being retrospectively designed, which obviously affect the overall value of the paper. Some of the baseline characteristics and results are too small for statistical analysis. Due to this reason, I suggest removing the "significantly" word in line 314.

Thank you for this comment. We have adjusted accordingly (line 357).

Reviewer 2 Report

This paper describes a retrospective observational study on the effects of maternal SSRIs on the immediate outcomes of preterm neonates.

1.  Lines 22-24, 122: since term infants were identified in the research for the study, it would be useful to compare their results to the preterm groups, with appropriate term controls.  This would also confirm that there were effects of maternal SSRIs in the term infants in this dataset.

2.  Add a hypothesis to the end of the Introduction.

3.  Give the p-values for the statistical comparisons in Table (similar to Tables 2 and 3).

4.  For all data, check the consistency in the number of decimal places in the mean and SD values, within each variable.

5.  Line 174-176: reverse the sentence to compare the exposed group to the non-exposed group.

6.  Line 182: delete the apostrophe in 'its'.

7.  Line 238: state which group is which for these data.

8.  Line 293-294: include power calculations to indicate whether the sample size was sufficient.

Author Response

Thank you for your comments. We greatly appreciate the feedback and have attempted to make the appropriate adjustments as possible.

Point 1 - Lines 22-24, 122: since term infants were identified in the research for the study, it would be useful to compare their results to the preterm groups, with appropriate term controls. This would also confirm that there were effects of maternal SSRIs in the term infants in this dataset.

Thank you for this important suggestion. We have intentionally designed this study to include only preterm infants as a control group, because we felt it would lower some of the bias risks due to differences between term and preterm infants. Furthermore, our dataset only included term infants who were admitted to NICU and did not include infants from the well-baby nursery. These infants were admitted to NICU for various reasons e.g respiratory distress, infection, hypoxic ischemic encephalopathy, and received disease specific individualized care, therefore we felt this heterogeneous population may not be appropriate as an additional control group.  Since much data have been collected in previous reports on term infants we chose to exclude them from this study. In order to acquire this information and add it to our dataset now we would need approval of the ethics committee and an extended timeline for resubmission of at least 2-3 months for this data acquisition, interpretation and revision of the manuscript accordingly. This could render this effort impractical. However, we added this point as another limitation of our study in the discussion section (lines 322-325).   

Point 2 - Add a hypothesis to the end of the Introduction.

Thank you for this important comment. We have now added a hypothesis (lines 80-82).

Point 3 - Give the p-values for the statistical comparisons in Table (similar to Tables 2 and 3).

Thank you for this comment. We have adjusted accordingly.

Point 4 - For all data, check the consistency in the number of decimal places in the mean and SD values, within each variable.

Thank you for this comment. We have adjusted accordingly.

Point 5 - Line 174-176: reverse the sentence to compare the exposed group to the non-exposed group

Thank you for this comment. We have adjusted accordingly (lines 188-189).

Point 6 - Line 182: delete the apostrophe in 'its'.

Thank you for this comment. We have adjusted accordingly (line 196).

Point 7 - Line 238: state which group is which for these data.

Thank you for this comment. We have adjusted accordingly (line 253).

Point 8 - Line 293-294: include power calculations to indicate whether the sample size was sufficient.

Thank you for this comment. We have added our sample size calculation with our baseline assumptions on page 3 lines 117-127. We have also discussed this sample size calculation in our discussion of our study limitations addressing some of the assumptions entered into the sample calculation that turned to be wrong (lines 310-318). We must always bear in mind the limitations of sample size calculations in retrospective studies.

Round 2

Reviewer 2 Report

The authors have addressed my comments.